



# 1 Intense *p*CO$_2$ and [O$_2$] Oscillations in a Mussel-Seagrass Habitat:
# 2 Implications for Calcification.

**4 Vincent Saderne[1,2], Peer Fietzek[1,3], Jens Daniel. Müller[4], Arne Körtzinger[1,5] and**
**5 Claas Hiebenthal[1]**

[1]{GEOMAR Helmholtz Centre for Ocean Research Kiel, Kiel, Germany}
[2]{KAUST King Abdullah University of Science and Technology, Thuwal, Kingdom of Saudi
Arabia}
[3]{Kongsberg Maritime Contros GmbH, Kiel, Germany}
[4]{Leibniz Institute for Baltic Sea Research, Warnemünde, Germany}
[5]{Christian Albrecht University, Kiel, Germany}
Correspondence to: V. Saderne (vincent.saderne@KAUST.edu.sa)

## 14 Abstract

Numerous studies have been conducted on the effect of ocean acidification on calcifiers inhabiting
nearshore benthic habitats, such as the blue mussel *Mytilus edulis*. The majority of these
experiments was performed under stable CO$_2$ partial pressure (*p*CO$_2$), carbonate chemistry and
oxygen (O$_2$) levels, reflecting present or expected future open ocean conditions. Consequently,
levels and variations occurring in coastal habitats, due to biotic and abiotic processes, were mostly
neglected, even though these variations largely override global long-term trends. To highlight this
hiatus and guide future research, state-of-the-art technologies were deployed to obtain high-
resolution time series of *p*CO$_2$ and [O$_2$] on a mussel patch within a *Zostera marina* seagrass bed,
in Kiel Bay (western Baltic Sea) in August and September 2013. Combining the in situ data with
results of discrete sample measurements, a full seawater carbonate chemistry was derived using
statistical models. An average *p*CO$_2$ more than 50% (~640 µatm) higher than current atmospheric
levels was found right above the mussel patch. Diel amplitudes of *p*CO$_2$ were large: 765 ± 310
(mean ± SD). Corrosive conditions for calcium carbonates ($\Omega_{arag}$ and $\Omega_{calc} < 1$) centered on sunrise

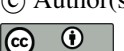


were found, but the investigated habitat never experienced hypoxia throughout the study period. It
is estimated that mussels experience conditions limiting calcification for 12 - 15 h per day, based
on a regional calcium carbonate concentration physiological threshold. Our findings call for more
extensive experiments on the impact of fluctuating corrosive conditions on mussels. We also stress
the complexity of the interpretation of carbonate chemistry time series data in such dynamic coastal
environments.


## 36    1      Introduction

Since preindustrial times, the atmospheric $CO_2$ mixing ratio rose from approximately 280 ppmv
to actual ~400 ppm (Mauna Loa, annual mean 2016, NOAA-ESRL). Future climate scenarios
predict a strong further increase of up to 1000 ppm by the year 2100 (IPCC, 2013). The dissolution
of anthropogenic $CO_2$ in seawater causes an increase in the seawater $CO_2$ partial pressure ($pCO_2$)
and a concurrent decrease of the seawater pH, a global phenomenon also referred to as ocean
acidification (OA) (Bates et al., 2014 and Doney et al., 2009).
OA reduces the supersaturation of seawater with respect to calcite and aragonite, which may cause
seawater to become corrosive for these calcium carbonates, that compose the shells and skeletons
of marine species (see Harvey et al., 2013 and Andersson et al., 2011 for review and meta-
analysis). In their review, Wahl et al. (2015) counted a total of 350 studies on the response of
benthic marine fauna and flora to ocean acidification, either in the field or in mesocosms. The
majority of these studies considered open ocean pH and $pCO_2$ as reference for the scenarios
employed to study the impact of OA. Indeed, the carbonate system (pH, $pCO_2$, dissolved inorganic
carbon (DIC) and total alkalinity (TA)) in surface oceans is understood relatively well, and many
models exist predicting the future rise of $pCO_2$ and decrease of pH (e.g. Orr et al., 2005). However,
very recent articles highlight the inapplicability of these predictions for the nearshore environment
(Duarte et al., 2013; Wahl et al., 2015; Müller et al., 2016) and the serious lack of relevant datasets
for the different major types of benthic habitats worldwide (Wahl et al., 2015). In addition to OA,
ocean warming and eutrophication of coastal waters around the world cause a spreading and
shoaling of hypoxia ($[O_2] < 60$ µmol kg$^{-1}$) in the ocean's interior (Diaz and Rosenberg, 2008;
Keeling et al., 2010).





In nearshore areas, metabolic processes by flora and fauna (photosynthesis, respiration and calcification) and redox reactions in the sediment (e.g. sulfate reduction/sulfide oxidation, denitrification/nitrification) strongly affect the carbonate chemistry of the water column, temporarily generating strong disequilibria for $CO_2$ and $O_2$ between the sea and the atmosphere. Cycles of super- and undersaturation for these two metabolic gases have been observed in all kinds of benthic habitats at daily and seasonal scale, but also in relation to physical forcing such as tides, wind or precipitation (Hofmann et al., 2011; Saderne et al., 2013; see Wahl et al., 2015 for review).

As an example, in the western Baltic Sea (Eckernförde Bay) Saderne et al. (2013) found daily variations of $p$CO$_2$ of 200 - 400 µatm between July and September in a macrophyte meadow dominated by the brown algae *Fucus serratus*, with maximum $p$CO$_2$ levels reaching up to approximately 2200 µatm during upwelling conditions. As a consequence, seawater saturation states for calcite and aragonite ($\Omega_{calc}$ and $\Omega_{arag}$) repeatedly fell below the dissolution threshold ($\Omega$ < 1, i.e. seawater turning undersaturated and hence corrosive to these biominerals) during the upwelling event, which was monitored over several days.

Here, we present a case study of a mixed seagrass/mussel bed, illustrating the carbonate chemistry conditions typically experienced by calcifiers in their natural habitat. We measured $p$CO$_2$ and dissolved oxygen ($[O_2]$) directly above a mussel patch of less than 2 m² extension, surrounded by seagrass, in a mosaic seagrass-mussel habitat as it is characteristic for western Baltic Sea nearshore benthic habitats. We used a combination of autonomous in situ sensors for $p$CO$_2$, $[O_2]$, salinity and temperature, complemented by discrete sampling for DIC, TA, phosphate, and silicate, for a period of more than 7 weeks in summer 2013. Using statistical modeling, we derived time series for the entire study period for carbonate chemistry parameters including $\Omega_{calc}$ and $\Omega_{arag}$. Based on literature knowledge regarding effects of carbonate ion concentration ($[CO_3^{2-}]$) as well as $\Omega_{calc}$ and $\Omega_{arag}$ effects on mussel biomineralization, we identified time windows of potentially favorable and unfavorable conditions for mussel calcification.

## 2. Materials and Methods

### 2.1 Description of Kiel Bay and the experimental site





Kiel Bay is a narrow and shallow bay in the western Baltic Sea that consists of two basins. The
inner Kiel Bay (see Fig. 1A) in the south is up to approximately 2 km wide and (except for a few
deeper dips) up to 14 m deep, while the outer Kiel Bay is up to approximately 5 km wide and
approximately 20 m deep and opens in the north to the larger Kiel Bight (Kögler and Ulrich, 1985;
Schwarzer and Themann, 2003). Surface and bottom water bodies of the larger Kiel Bight swash
in and out Kiel Bay (mainly driven by varying wind directions and intensities). Freshwater influx
from the small river "Schwentine" (east side of Kiel Bay, opposite to GEOMAR on Fig. 1A) also
shapes the hydrology of the inner Kiel Bay. The strict separation of deep and shallow water bodies
by stratification in the larger Kiel Bight and its bays that typically occurs during summer months
is occasionally broken-up by wind-driven upwelling. The shallow coastal communities are then
put in contact with high $CO_2$ and sometimes hypoxic sub-surface waters (Feely et al., 2008;
Melzner et al., 2013).
In Kiel Bay, as in several other enclosed bays of the western Baltic Sea, the blue mussel *Mytilus*
*edulis* and the seagrass *Zostera marina* co-occur in patches forming mosaic habitats (Reusch and
Chapman, 1995; Vinther et al., 2008; Vinther and Holmer 2008; Vinther et al. 2012). A sensor
package measuring $p$CO$_2$, [O$_2$], salinity and temperature was deployed at 2 m depth in a mixed
habitat formed by *Z. marina* and *M. edulis* in Kiel Bay, western Baltic Sea (54.3467 °N,
10.1539 °E; see Fig.1A). The package was directly placed on a mussel patch within the seagrass
bed. The deployment was conducted for 50 days from the 08.08.2013 to 27.09.2013 with short
power interruptions from 10.08., 17:10 to 11.08., 8:00 and from 12.08., 5:00 to 14.08., 16:00.
**2.2   *In situ* sensor suite**
Temperature, salinity and [O$_2$] were measured simultaneously every 10 min with a SBE 37–SI
MicroCAT (temperature and salinity, Sea-Bird Electronics Inc., USA) and an oxygen optode
Aanderaa 3835 (Aanderaa Data Instruments AS, Norway) enclosed in a flow cell. The circulation
of water between the SBE 37–SI and the optode was achieved by means of an SBE 5M pump
(Sea.-Bird Electronics Inc., USA) that ran for 30 seconds every 10 min. The coordination of
pumping and recording by the SBE 37 and the optode was carried out by a custom-made data
logger (Todd Martz Laboratory, Scripps Institution of Oceanography, San Diego, USA). To
prevent fouling on sensors, the SBE 37–SI was equipped with tributyltin tablets and copper tubing
linked the SBE 37–SI and the flow cell to the pump.



A HydroC® $CO_2$ II sensor (KM Contros GmbH, Kiel, Germany) was used to autonomously
measure in situ $CO_2$ partial pressure ($pCO_2$). The sensor determines $pCO_2$ optically by means of
an nondispersive infrared (NDIR) absorption measurement within a membrane-equilibrated
headspace (Fietzek et al., 2014). The sensor was calibrated at a water temperature of 17.5 °C at 6
different $pCO_2$ levels across a measurements range of 200 - 2200 µatm by the manufacturer before
(July) and after (November) the measurements. The corresponding calibration polynomials had a
quality of $R^2$ = 0.999998 and 0.99998 with root mean square errors of 1.15 and 3.98 ppm.
Calibrations and data processing were carried out according to Fietzek et al. (2014) with the
exception that, here, the interpolation between the pre- and the post-deployment calibration
polynomial was carried out according to the sensor's absolute run-time between the two
calibrations.
During the field deployment, the sensor was powered from the nearby pier. Data were stored
internally on its data logger. The sensor was configured to carry out a 2 min zeroing every 6 h. A
flushing interval duration of 55 min was used to analyze the data during recovery from zero to
ambient values. During the subsequent measurement interval, a 10 s mean of the 1 Hz raw data
was stored every minute.
The HydroC® was equipped with a flow-head allowing for passive diffusion of seawater to the
sensor's membrane through a circumferential orifice along the cylindrical sensor housing. In order
to improve the data quality under the given configuration and deployment conditions,
determination and correction of the sensor's response time (RT) were given special attention (see
Appendix A for further information). The sensor's in situ RT was determined to be 546 ± 208 s
(mean ± SD) and the final $pCO_2$ series was response time corrected assuming a constant RT of
546 s.
We conclude a general uncertainty for the final $pCO_2$ series in this study of 2.5% of reading as the
standard deviation of the $pCO_2$ measurements. This value is comprised of the accuracy of drift
corrected HydroC® $pCO_2$ data of approximately 1% of reading as found within Fietzek et al. (2014)
and the uncertainty estimate of 1.5% of reading related to the actual RT influences (see Appendix
A).
**2.3   Discrete sampling**



Over the course of the deployment, a total of 31 seawater samples for DIC and TA were taken at
the sensor suite through snorkeling. Sampling was conducted twice a week in the hours following
sunrise and solar noon. On a third day, duplicate sampling was conducted in the hour following
solar noon. Corresponding sampling results were averaged to improve the quality of the
measurements. Salinity of the water samples was measured in a laboratory at GEOMAR using a
conductometer (SG 7/8, Mettler Toledo, Switzerland). Subsequently, the samples were poisoned
with mercury chloride following the recommendations by Dickson et al. (2007). DIC (precision ±
3 µmol kg$^{-1}$) was measured by coulometry using a SOMMA instrument (University of Rhode
Island, USA) and TA (precision ± 6 µmol kg$^{-1}$) was determined with a VINDTA titrator (Marianda
GmbH, Germany) following Dickson et al. (2007).
In parallel to all DIC and TA samplings, another set of seawater samples was taken and frozen for
measurement of phosphate and silicate concentrations. Total phosphate (precision 0.1 µmol kg$^{-1}$)
and total silicate (precision 0.2 µmol kg$^{-1}$) concentrations were measured using a QuAAtro auto-
analyzer with an XY-2 sampler (SEAL Analytical GmbH, Germany).

### 2.4.1  Correction for Organic Alkalinity

The $p$CO$_2$ in the discrete sample was calculated from measured DIC, TA, total phosphate and total
silicate using the first and second carbonate system dissociation constants for brackish waters from
Millero (2006) and the dissociations constants of HF and HSO$_4^-$ of Perez and Fraga (1987) and
Dickson (1990), respectively, with the R package Seacarb (Lavigne and Gattuso, 2013).
A critical point for the calculation of carbonate chemistry in waters containing significant amounts
of dissolved organic matter, in the following referred to as DOC, is the contribution of organic
acid-base components to the TA (Cai et al., 1998; Kuliński et al., 2014 and Yang et al., 2015). This
organic TA contribution (TA$_{org}$) is not reflected in models employed to interpret titration data nor
in equations routinely used to perform carbonate system calculations. The classical concept, i.e.
two out of four measureable carbonate system parameters are sufficient to calculate the remaining,
is limited if TA is one of the measured parameters and the sample contains high amounts of DOC.
In such cases, the TA value determined by titration can significantly exceed the amount of TA
contributed by the inorganic acid-base components. This hinders an accurate quantification of the
inorganic alkalinity and thereby affects the calculation of other carbonate system parameters.



Kuliński et al. (2014) demonstrated that the $p$CO$_2$ calculated from TA and DIC is typically 100 -
200 µatm lower than the measured $p$CO$_2$ in open waters of the Baltic Sea. This deviation is not
observed if $p$CO$_2$ is calculated from measured DIC and pH data (Fig. 2), which are unaffected by
the TA$_{org}$ contribution. Two aspects of the present study therefore require the consideration of
TA$_{org}$: (i) TA$_{org}$ explains the observed differences between $p$CO$_2$ measured in situ and $p$CO$_2$
calculated from DIC, TA, silicate and phosphate of discrete samples (Fig. 2) and (ii) TA$_{org}$ needs
to be considered when other carbonate system parameters (pH$_T$, DIC, $\Omega_{arag}$ and $\Omega_{calc}$) are calculated
from the TA and $p$CO$_2$ time series.
The offset between measured and calculated $p$CO$_2$ caused by TA$_{org}$ increases towards higher $p$CO$_2$
levels. This could be repeatedly observed during another measuring campaign in Kiel Bay
(Hiebenthal et al., 2017, Fig. 2). To furthermore unravel the difference in observed versus
calculated $p$CO$_2$ for typical Kiel Bay conditions (S = 16, T = 18 °C, DOC = 300 µmol kg$^{-1}$, TA =
1950 µmol kg$^{-1}$), we qualitatively reproduced the impact of TA$_{org}$ on the carbonate system by a
modelling approach using regional TA$_{org}$ properties reported by Kuliński et al. (2014). Our
modelling approach revealed an offset between measured and observed $p$CO$_2$ that increases with
CO$_2$ partial pressure and reaches up to 300 µatm at $p$CO$_2$ levels around 2000 µatm (Fig. 2, solid
line). A more detailed description of the modelling approach is given in Appendix B.
For the calculation of carbonate system parameters (pH, DIC, $\Omega_{arag}$ and $\Omega_{calc}$) from TA and $p$CO$_2$
data, we consequently corrected the TA time series originally based on titration measurements for
a TA$_{org}$ contribution. Therefore we used 18 discrete seawater samples taken at 1 m depth right next
to a constantly deployed HydroC$^®$ CO$_2$ sensor at GEOMAR pier in the inner Kiel Bay, 2 km south
of the above described experimental site between March and December 2015 (Hiebenthal et al.,
2017). The water samples were poisoned with HgCl$_2$ (Dickson et al., 2007) within 15 min and
stored until measurement of DIC, pH$_T$ and TA at the Leibniz Institute for Baltic Sea Research,
Warnemünde, Germany. DIC was analyzed with a SOMMA system at 15 °C. TA was determined
by an open-cell titration at 20 °C. Certified reference materials provided by Andrew Dickson's
laboratory were measured in parallel for quality assurance. The pH$_T$ of each water sample was
determined spectrophotometrically at 25 °C with unpurified m-cresol purple as indicator dye
(Hammer et al., 2014). Phosphate and silicate concentrations used for the 2015 samples were

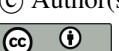



measure after Grasshoff 1999 and sampled approximately 240 m from GEOMAR pier (Hiebenthal
et al., 2017).
The $TA_{org}$ fraction measured in Kiel inner Bay, determined as the difference between TA measured
and TA calculated from DIC and $pH_T$, at GEOMAR pier 2015 was 0.84 ± 0.0005% (mean ± SE),
corresponding to a contribution between approximately 8 and 30 µmol $kg^{-1}$. We chose to consider
this conservative range for the carbonate chemistry calculations instead of the values determined
during the 2013 deployment, 0.48 ± 0.28% (mean ± SE). The reason for using the 2015 value is
that it was determined from measurements of TA, DIC and $pH_T$ in the same water samples
(Hiebenthal et al., 2017), thereby avoiding the additional uncertainty and noise in the data due to
a spatio-temporal mismatch between $p$CO₂ sensor and discrete sample data of the 2013 field
deployment.

## 2.5    Seawater carbonate chemistry

We used a model approach based on discrete water sample- and salinity data to estimate total
alkalinity (TA) from salinity (S). Given the obvious changes particularly in phosphate and salinity
around September 1st (Fig. 3A-B), two separate TA-S regressions were calculated for August and
September (Fig. 4, Tab. 1). The slopes and intercepts were used to derive total alkalinity from the
salinity time series. Both regressions were highly significant with p-values < 0.001, $R^2 = 0.64$ and
0.92 for August and September respectively and standard deviations of the residuals < 15 µmol
$kg^{-1}$ (Fig. 4). The two intercepts for August and September are notably different by 100 µmol $kg^{-1}$
while the two slopes are similar (~40 µmol $kg^{-1}$). An organic-free TA range was calculated
($TA_{inorg}$) by subtracting constant organic alkalinity contributions of 8 and 30 µmol $kg^{-1}$ (lower and
upper $TA_{org}$ limits in 2015 measurements at GEOMAR pier; see 2.4) from the TA time series.

## 2.6    Calculation of the regional atmospheric $p$CO₂

Half-hourly measured CO₂ mole fractions in dry air from the German Federal Environment
Agency (Umweltbundesamt), Station Westerland, 8.3082 °E and 54.9250 °N, were averaged for
the months August (08th – 31st 2013) and September (01st – 27th 2013): 391 ± 7 and 395 ± 14 ppm
(mean ± SD). Thereof $p$CO₂ in wet air (100% relative humidity at SST) of 385 and 388 µatm for
August and September, respectively, were derived at local measurement conditions; i.e. using an
averaged sea surface temperature (18.4 ± 0.6 and 16.1 ± 1.0 °C) and ambient pressure readings





(1019.6 ± 4.2 and 1015.7 ± 7.5 mbar; both parameters from GEOMAR meteorological station,
Fig. 1A) as well as the salinity measured in this study (15.6 ± 0.7 and 15.9 ± 1.0).
## 2.7    Inferential statistics
Daily means, maxima, minima as well as day-night peak-to-peak amplitudes and daily duration of
undersaturation for calcite and aragonite (in hours) (mean ± SD) for the months of August and
September were compare using Mann-Whitney U tests. Statistical analyses were conducted with
Statistica 7 (Statsoft, USA).
# 3    Results
## 3.1    Salinity, water temperature, total silicate and total phosphate observations
Salinity did not show circadian patterns and the daily mean salinity was not significantly different
between August and September (Mann-Whitney U, $p > 0.05$, see Appendix C for detailed
statistics), with 15.65 ± 0.7 and 15.89 ± 1.01 (mean ± SD), respectively (Fig. 3A). The total
alkalinity (TA), derived from salinity, varied between 1934 and1956 µmol kg$^{-1}$ during the 2
months, with a SD of 41 µmol kg$^{-1}$. As a marker of seasonality, temperature daily means, minima,
maxima were all significantly lower in September compared to August (Mann-Whitney U, $p <$
0.001, see Appendix C), with drops of 2.3 °C, 2.2 °C and 3 °C, respectively (Tab. 2, Fig. 3A, 4A).
The daily variation amplitude was significantly reduced in September by 0.3°C compared to
August (Mann-Whitney U, $p < 0.01$, see Appendix C). Total silicate was not significantly different
between August and September: 19.0 ± 4.9 µmol kg$^{-1}$ and 18.8 ± 5.4 µmol kg$^{-1}$ (Mann-Whitney
U, $p > 0.05$, see Appendix C). Total phosphate was significantly different between August and
September: 0.7 ± 0.14 µmol kg$^{-1}$ and 5.9 ± 3.7 µmol kg$^{-1}$ (Mann-Whitney U, $p < 0.001$, see
Appendix C).
## 3.2    $p\mathrm{CO_2}$ and [$\mathrm{O_2}$] observations
The daily mean (± SD) $p\mathrm{CO_2}$ of 628 ± 114 µatm in August and 652 ± 193 µatm in September (Tab.
2) remained always above the regional atmospheric $p\mathrm{CO_2}$ of approximately 387 µatm. The mean
daily minimum and maximum $p\mathrm{CO_2}$ values were 334 ± 119 µatm and 1151 ± 328 µatm in August
and 373 ± 139 µatm and 1097 ± 336 µatm (mean ± SD) in September, respectively (see Tab. 2,




Figs. 3C, 5B). A modest and non-significant increase of the daily means was observed between
August and September (+ 22 µatm) as well as a non-significant average decrease of the day-night
variability by 93 µatm (Mann-Whitney U, p > 0.05, see Appendix C) (Tab. 2). A high $p$CO$_2$ event
was observed between September 8$^{th}$ and September 12$^{th}$ with a peak of the daily mean $p$CO$_2$ to
1166 µatm on September 9$^{th}$. On this day, a maximum $p$CO$_2$ of 1839 µatm was observed at
04:30 a.m. (see Fig. 3C).
The daily mean [O$_2$] most of the time remained below the seawater saturation threshold (~260 to
290 µmol kg$^{-1}$) (Fig. 3D, Tab. 2). However, supersaturation due to photosynthesis was regularly
observed between noon and sunset in August. Significant decreases of the daily average, maximum
and minimum [O$_2$] by 47 µmol kg$^{-1}$, 64 µmol kg$^{-1}$, 35 µmol kg$^{-1}$, respectively, were observed in
September (Mann-Whitney U, all p < 0.001, see Appendix C for details) (Tab. 2, Fig. 5C), with
an abrupt [O$_2$] decrease occurring on Sept. 8$^{th}$, in parallel to a sudden decrease in temperature by
approximately 2 °C (Fig. 3A-D). A significant decrease of the day-night amplitude by 29 µmol kg$^{-1}$
was observed from August to September (Mann-Whitney U, p = 0.008, Appendix C). The
threshold for hypoxia (60 µmol kg$^{-1}$) is never undercut (see Fig. 3D); the minimum daily mean
concentration observed during the two months was 140 µmol kg$^{-1}$.

## 3.3    Derived carbonate chemistry parameters

Times series for pH$_T$, DIC, $\Omega_{arag}$ and $\Omega_{calc}$ (Fig. 6) were derived from modeled TA$_{inorg}$ (Fig. 4) and
measured $p$CO$_2$ and analyzed for daily means, minima, maxima and diel peak-to-peak amplitudes
(mean ± SD) with TA$_{org}$ = 8 µmol kg$^{-1}$ and TA$_{org}$ = 30 µmol kg$^{-1}$ (Tab. 1). Since the differences in
the calculated parameters between the TA$_{org}$ = 8 µmol kg$^{-1}$ and the TA$_{org}$ = 30 µmol kg$^{-1}$ estimates
are too small to be recognizable in Fig. 6, only results for the mean TA$_{org}$ = 19 µmol kg$^{-1}$ are
shown. Overall, we observe a slight and non-significant (Mann-Whitney U, p > 0.05, see Appendix
C for details) decrease in daily means of DIC and pH$_T$ between August and September by 11 µmol
kg$^{-1}$ and 0.02 pH$_T$ units respectively (Tab. 1). In parallel, we observe a non-significant decrease of
the amplitudes of the diel variations in DIC of 21 µmol kg$^{-1}$ (Mann-Whitney U, p > 0.05, see
Appendix C) and pH$_T$ of approximately 0.05 pH$_T$ units (Mann-Whitney U, p > 0.05, see Appendix
C) and (Tab. 1, Fig. 5). All these observations are coherent with the changes in $p$CO$_2$ previously
described.



Daily means of $\Omega_{arag}$ are close to the saturation threshold: between 1.3 and 1.4 in August and 1.2
in September (Tab. 1). For both, $\Omega_{arag}$ and $\Omega_{calc}$, we observe a significant decrease in daily mean
values (Mann-Whitney U, all p < 0.01, see Appendix C) and a marginally significant decrease of
diel amplitudes between August and September (Mann-Whitney U, all p < 0.1, see Appendix C)
(Tab. 1, Fig. 5). In both isoforms, the amplitudes decrease results from a significant reduction of
the daily maxima (Mann-Whitney U, p = 0.001, see Appendix C) with the minima remaining
constant (Mann-Whitney U, p > 0.05, see Appendix C) (Tab. 1, Fig. 5). On average, the seawater
was undersaturated with respect to aragonite for approximately 6 hours per days in August and
approximately 9 hours per day in September (Tab. 1). Similarly, seawater was undersaturated with
respect to calcite for approximately 30 min and approximately 1 hour 30 min in August and
September, respectively (Tab. 1). Only one full day of undersaturation with respect to aragonite
was observed on September 9th during the high $p$CO$_2$ event (Fig. 5).
Over the whole measurement period, the consideration of organic alkalinity contributions has little
effect on the derived CO$_2$ system parameters (mean value differences: 20 µmol kg$^{-1}$ DIC, < 2 µmol
kg$^{-1}$ [CO$_3^{2-}$] and negligible effect on omegas and pH$_T$). The only noticeable difference when taking
TA$_{org}$ into account was an increase of time of undersaturation for $\Omega_{arag}$ and $\Omega_{calc}$ in September of
approximately 25 min and approximately 10 min, respectively.
**4    Discussion**
**4.1    $p$CO$_2$, carbonate chemistry and O$_2$**
Monthly averages of $p$CO$_2$ close to the seafloor as presented in this study (~640 µatm; Tab. 1)
were more than 50% above atmospheric $p$CO$_2$. In 2011, Saderne et al. (2013), used similar
technologies as in our study in a seaweed dominated bed of Eckernförde Bay (adjacent to Kiel
Bay, Western Baltic) and found significantly lower weekly mean $p$CO$_2$ values (~390 µatm in July,
~240 µatm in August and 420 µatm in September, excluding an upwelling event, Saderne et al.,
2013). Accordingly, the day-night amplitudes of $p$CO$_2$ observed in the present study (764 ± 310
µatm, overall mean ± SD) are 3 to 4 times higher than observed during "normal" days by Saderne
et al. (2013) (243 ± 95 µatm) in the neighboring Eckernförde Bay in 2011. This reflects a
characteristic of the marine carbonate system: Equal DIC variations will induce stronger $p$CO$_2$
variations at high "baseline" $p$CO$_2$ levels compared to lower $p$CO$_2$ levels. The source of this pattern





is the reduced buffering capacity of the carbonate system at a high $p\text{CO}_2$ "baseline". Accordingly,
Saderne et al. (2013) found extreme $p\text{CO}_2$ variations of approximately 1700 µatm during an
upwelling event that lifted the "baseline" $p\text{CO}_2$ to approximately 1600 µatm, although the DIC
variations due to plant photosynthesis remained rather unchanged by the upwelling. Likewise, the
amplitude of the diel DIC variations during our study (145 µmol kg$^{-1}$ in August and 124 µmol kg$^{-1}$
in September, see Tab.1 and Fig. 5) were of the same magnitude as what had been observed in
Eckernförde Bay before and after upwelling (141 µmol kg$^{-1}$ and 106 µmol kg$^{-1}$,
respectively,Saderne et al., 2013).
Average $\text{O}_2$ concentrations were below saturation in August and September 2013, with a
significant decrease occurring in September (monthly means of 89.4% $[\text{O}_2]_{\text{sat}}$ and 68.8% $[\text{O}_2]_{\text{sat}}$
for August and September respectively). However, we note that at no point of our survey the
threshold of hypoxia (~22% saturation; 60 µmol kg$^{-1}$) was reached.The significant $\text{O}_2$ decrease in
September co-occurred with an increase in phosphate and a change in DIC to TA regression. This
can possibly be explained by a rapid degradation of the seagrass shouts observed in September and
a possible shift of the habitat from an auto- to a heterotrophic system.

## 4.2    Implications for mussel calcification in a seagrass meadow

We found pronounced variations of $\Omega_{\text{calc}}$ and $\Omega_{\text{arag}}$ on a daily basis, e.g., resulting in 5.7 to 8.8 h
of undersaturation for aragonite per day in the water body right above the mussel patch.
As expected, daytime photosynthesis counters water corrosiveness caused by heterotrophic
processes, while at night water corrosiveness is reinforced by respiration. Waldbusser et al. (2014)
demonstrated that saturation states (and therefore $[\text{CO}_3^{2-}]$) are the parameters affecting the larval
development and growth of *Mytilus galloprovincialis* and *Crassostrea gigas*. On young *M. edulis*
Hiebenthal et al. (2013) found a negative correlation between growth and $\Omega$ or $[\text{CO}_3^{2-}]$. Thomsen
et al. (2015) confirmed these findings in larvae and juveniles. With a meta-analysis including all
past work on mussel populations from Kiel Bay, they found that the critical $\text{CO}_3^{2-}$ concentration
below which calcification starts to decline was 80 µmol kg$^{-1}$ (although they specified that the
directly related ratio $[\text{H}^+]/[\text{HCO}_3^-]$ is likely to be the controlling parameter for calcification,
Thomsen et al. 2015). In our survey, mussels were exposed to $[\text{CO}_3^{2-}]$ below this threshold for 12
$\pm$ 5.2 h per day in August and 15.3 $\pm$ 5.4 h per day in September (mean $\pm$ SD). Comparing these





findings to the shorter durations of aragonite (5-9 h per day) and calcite (~1 h per day)
undersaturation implies that reduced calcification rates might already occur during periods with
low, yet oversaturated calcium carbonate concentrations. However, the consequences of these
successions of intense corrosive stress and stress relaxation on the juvenile and adults forming the
mussel patch is still under debate. On mussel larvae, Frieder et al., 2013 showed that the negative
effects of elevated $p$CO$_2$ (~1500 µatm) on *Mytilus galloprovincialis* disappear, if diel variations
of 500 µatm were added, although this effect was not observed for *Mytilus californianus*. Wahl et
al. (2017) found that in laboratory and mesocosm experiments calcification of blue mussels is
significantly higher during daytime, when photosynthetic activity of macrophytes creates
favorable calcification conditions. However, in field studies they did not detect a positive net effect
of the co-occurrence with macrophytes on the calcification of mussels (Wahl et al. 2017).
Furthermore, Thomsen et al. (2013) showed that high food availability, particularly in Kiel Bay,
can circumvent the effects of acidification in mussels.

**Conclusion**

Our study demonstrates how essential it is to place more effort in measuring the carbonate
chemistry variations in nearshores habitats, and highlights the need to include variability when
investigating the impact of OA on benthic organisms. We emphasize that continuous carbonate
system observations in benthic habitats are possible but challenging due to the high spatio-
temporal variability and organic alkalinity contributions. However, with the here applied
combination of in situ sensor measurements, laboratory analyses of discrete water samples, and
modelling elements we were able to distinguish daily oscillations and shifts of averages (across
weeks) of several seawater chemistry parameters. This approach specifically allowed for a
temporal quantification of dissolution threshold undercuts *M. edulis* experiences in this exemplary
site of Kiel Bay.

**Appendix A: HydroC® response time and related signal processing**

The entire HydroC® $p$CO$_2$ time series has a total of 187 zeroings and related flush intervals. A first
order kinetics model was fitted to every of the HydroC®'s signal recoveries from its zero value to
ambient partial pressure over an 55 min flush interval to obtain sensor response times (see Fiedler



et al. 2013). The fit interval was set to extend over 55 min to be around 6 times as large as the
sensor's response time to allow for reasonable fitting of the exponential increase. In general the
carbonate system at the site was characterized by a strongly varying baseline featuring very steep
$pCO_2$ gradients over the course of the day with slopes of up to -54 µatm/min. Therefore the $pCO_2$
signal recovery from zero to ambient during the 187 flush intervals was often superimposed with
a changing background partial pressure. These adverse conditions hamper the response time
determination by means of a first order kinetics fit. We therefore flagged and not considered further
the recoveries providing the worst fit results, which indicates a bad match between model and real
signal: (i) the fits characterized by the largest 10% of root mean square (RMS) residuals of the
fitted curve and the real signal as well as (ii) the fits with the largest 10% of the uncertainty of the
fitted response time.
Finally a total of 157 fit results were considered providing an average response time ($t_{63}$) of
$546 \pm 208$ s with an RMS of the fit function of $6.0 \pm 3.1$ µatm and a fit uncertainty for the response
time of $1.2 \pm 0.4$ s. The average response time found is in good agreement with a $t_{63}$ of
$553 \pm 8$ µatm as determined during a dedicated laboratory test with a similar sensor setup at 14.5°C
water temperature. The large standard deviation of the averaged in situ response times (i.e. 208 s)
is likely caused (i) by the influence of the strongly varying background $pCO_2$ on the data to be
fitted and (ii) by the variability of the water exchange in front of the membrane within the flow-
head as caused by i.e. changes in the ambient water currents. Against the large variability in the
determined response times, a temperature dependence of the $t_{63}$ can be neglected as well as a
potential response deceleration caused by fouling on the membrane, which was only observed to
a very small extend after the deployment.
The response time (RT) correction according to Miloshevich et al. (2004) and Fiedler et al. (2013)
was carried out with a constant response time of 546 s to obtain the final $pCO_2$ series. The $pCO_2$
data were additionally RT-corrected with a reduced ($RT_{short} = 546 - 208 = 338$ s) and an extended
response time ($RT_{long} = 546 + 208 = 754$ s). In order to estimate the uncertainty of the final $pCO_2$
time series, the differences between the $RT_{long}$- and the RT-corrected as well as between the RT-
and the $RT_{short}$-corrected $pCO_2$ time series were determined. They both provide a small average
$\Delta pCO_2$ of 0.8 µatm (0.06% of reading) and a corresponding standard deviation of 10 µatm (1.5%
of reading). The standard deviation is influenced by short periods characterized by large $pCO_2$
gradients where the RT-correction has the largest effect; i.e. at 130 µatm/min over 3 minutes




(54 µatm/min in the original, non-RT-corrected data) the steepest $pCO_2$ decline of the series was
observed in the early morning of August $22^{nd}$ at around 4:40 causing the maximum observed
individual $\Delta pCO_2$ of -172 µatm or 23% of reading.
The standard deviation of the $\Delta pCO_2$ (1.5% of reading) is used as a measure for the $pCO_2$
uncertainty in this study related to the response time influences. Not considering the response time
and applying a related correction, would have forced us to continue the analysis with a less
realistic, "smoothed" data set that showed temporally delayed as well as amplitude-damped $pCO_2$
peaks and troughs.
Fiedler, B., Fietzek, P., Vieira, N., Silva, P., Bittig, H. C. and Körtzinger, A.: In Situ $CO_2$ and $O_2$
Measurements on a Profiling Float, J. Atmos. Ocean. Technol., 30(1), 112–126,
doi:10.1175/JTECH-D-12-00043.1, 2013.
Miloshevich, L. M., Paukkunen, A., Vömel, H. and Oltmans, S. J.: Development and validation of
a time-lag correction for Vaisala radiosonde humidity measurements, J. Atmos. Ocean. Technol.,
21(9), 1305–1327, 2004.
**Appendix B**
We implemented a model to estimate the potential error associated to $CO_2$ system calculations, in
case one of the input parameters is titrated TA including organic acid-base components.
Hence, we firstly calculated the carbonate system at varying DIC and constant TA, salinity, and
temperature. In this case the derived carbonate system parameters (i.e. $pCO_2$) reflect values as
calculated from TA and DIC measured in discrete samples.
Organic contribution to TA ($TA_{org}$) is estimated based on the relation reported by Kuliński et al.
(2014) for the Baltic Sea, which approximates $TA_{org}$ from the proton concentration [$H^+$], the
amount of DOC, a bulk dissociation constant ($K_{DOM}$) and the fraction of DOC ($f$) acting as a carrier
of weakly acidic groups
$$TA_{org} = \frac{K_{DOM} \cdot f \cdot DOC}{[H^+] + K_{DOM}}$$
This $TA_{org}$ estimate is a side-specific approximation since $K_{DOM}$ and $f$ ($2.94 \times 10^{-8}$ mol $kg^{-1}$ and
0.14, respectively) are characteristic for the actual DOM composition. Further, $K_{DOM}$ is currently
only reported for T=25°C and no salinity-dependence was investigated. Therefore, this model is
only a qualitative approximation of the carbonate system in high DOC waters.



The inorganic TA fraction can be approximated as $TA_{inorg} = TA - TA_{org}$. This $TA_{inorg}$ together with
DIC can be used to calculate the "correct" $pCO_2$ ($pCO_{2,sensor}$) as it would also be obtained from
direct observations, since $TA_{inorg}$ represents the share of the alkalinity which is covered by the
dissociation constants and equilibrium reactions implemented in the routinely applied carbonate
system models (i.e. $CO_2$sys, seacarb).
Finally, the $TA_{org}$ estimate needs to be refined. Above, it was estimated from the proton
concentration, which itself was derived from TA and DIC. Now, $TA_{inorg}$ is the better input
parameter to calculate this proton concentration. Thus, we recalculate first $TA_{org}$ to get a refined
value ($TA_{org,ref}$). The other carbonate system parameters, including $pCO_{2,sensor,ref}$, are calculated
with $TA_{org,ref}$. This refinement procedure was repeated iteratively, until no significant changes
occurred between the iterative steps. The finally obtained $pCO_{2,sensor,ref}$ would reflect direct $pCO_2$
measurements performed with a dedicated in situ sensor.
Model output:
The model predicts an increase in the deviation between $pCO_2$ measured in situ ($pCO_{2,sensor,ref}$) and
$pCO_2$ values obtained by calculations from discrete sample TA and DIC towards $pCO_2$ levels
around 2000 µatm as depicted in Fig. 2 (input parameters: S = 16, T = 18 °C, DOC = 300 µmol
$kg^{-1}$, TA = 1950 µmol $kg^{-1}$, $K_{DOM}$ and $f$ from Kuliński et al. (2014), $CO_2$ system constants from
Millero et al. (2006)).
Kuliński, K., Schneider, B., Hammer, K., Machulik, U. and Schulz-Bull, D.: The influence of
dissolved organic matter on the acid–base system of the Baltic Sea, J. Mar. Syst., 132, 106–115,
doi:10.1016/j.jmarsys.2014.01.011, 2014.
Millero, F. J., Graham, T. B., Huang, F., Bustos-Serrano, H. and Pierrot, D.: Dissociation constants
of carbonic acid in seawater as a function of salinity and temperature, Mar. Chem., 100(1-2), 80–
94, doi:10.1016/j.marchem.2005.12.001, 2006.

**Appendix C: Mann-Whitney U tests details**
Result table for the Mann Whitney tests comparing August and September data listed in Table 2.

|  | U | Z | p-value | August n | September n |
|---|---|---|---|---|---|
| Mean Temperature | 2 | 5.71 | <0.01 | 20 | 26 |
| Min. Temperature | 5 | 5.64 | <0.01 | 20 | 26 |
| Max. Temperature | 1 | 5.73 | <0.01 | 20 | 26 |
| ΔTemperature | 133 | 2.80 | 0.01 | 20 | 26 |



| | | | | | |
|---|---|---|---|---|---|
| Mean Salinity | 198 | -1.36 | 0.17 | 20 | 26 |
| Min. Salinity | 196 | -1.41 | 0.16 | 20 | 26 |
| Max. Salinity | 186 | -1.63 | 0.10 | 20 | 26 |
| $\Delta$Salinity | 252 | -0.17 | 0.87 | 20 | 26 |
| Total Silicate | 114 | 0.100 | 0.920 | 13 | 18 |
| Total Phosphate | 0 | -4.665 | <0.001 | 13 | 18 |
| Mean $pCO_2$ | 255 | 0.10 | 0.92 | 20 | 26 |
| Min. $pCO_2$ | 220 | -0.88 | 0.38 | 20 | 26 |
| Max. $pCO_2$ | 233 | 0.59 | 0.56 | 20 | 26 |
| $\Delta pCO_2$ | 220 | 0.88 | 0.38 | 20 | 26 |
| Mean $[O_2]$ | 67 | 4.758 | <0.001 | 24 | 26 |
| Min. $[O_2]$ | 121 | 3.709 | <0.001 | 24 | 26 |
| Max. $[O_2]$ | 68 | 4.738 | <0.001 | 24 | 26 |
| $\Delta[O_2]$ | 176 | 2.641 | 0.008 | 24 | 26 |

| | | $TA_{org} = 8$ | | | |
|---|---|---|---|---|---|
| | U | Z | p-value | August n | September n |
| Mean $pH_T$ | 241 | 0.41 | 0.68 | 20 | 26 |
| Min. $pH_T$ | 246 | -0.30 | 0.76 | 20 | 26 |
| Max. $pH_T$ | 212 | 1.05 | 0.29 | 20 | 26 |
| $\Delta pH_T$ | 215 | 0.99 | 0.32 | 20 | 26 |
| Mean DIC | 254 | 0.12 | 0.90 | 20 | 26 |
| Min. DIC | 227 | -0.72 | 0.47 | 20 | 26 |
| Max. DIC | 203 | 1.25 | 0.21 | 20 | 26 |
| $\Delta$DIC | 198 | 1.36 | 0.17 | 20 | 26 |
| Mean $[CO_3^{2-}]$ | 172 | 1.94 | 0.05 | 20 | 26 |
| Min. $[CO_3^{2-}]$ | 225 | 0.76 | 0.44 | 20 | 26 |
| Max. $[CO_3^{2-}]$ | 153 | 2.36 | 0.02 | 20 | 26 |
| $\Delta[CO_3^{2-}]$ | 181 | 1.74 | 0.08 | 20 | 26 |
| Mean $\Omega_{arag}$ | 157 | 2.27 | 0.02 | 20 | 26 |
| Min. $\Omega_{arag}$ | 213 | 1.03 | 0.30 | 20 | 26 |
| Max. $\Omega_{arag}$ | 150 | 2.43 | 0.02 | 20 | 26 |
| $\Delta \Omega_{arag}$ | 176 | 1.85 | 0.06 | 20 | 26 |
| time $\Omega_{arag} < 1$ | 176 | -1.85 | 0.06 | 20 | 26 |
| Mean $\Omega_{calc}$ | 167 | 2.05 | 0.04 | 20 | 26 |
| Min. $\Omega_{calc}$ | 215 | 0.99 | 0.32 | 20 | 26 |
| Max. $\Omega_{calc}$ | 152 | 2.38 | 0.02 | 20 | 26 |



| | U | Z | p-value | August n | September n |
|---|---|---|---|---|---|
| Δ Ω$_{calc}$ | 178 | 1.81 | 0.07 | 20 | 26 |
| time Ω$_{calc}$ <1 | 212 | -1.05 | 0.29 | 20 | 26 |
| TA$_{org}$ = 30 | | | | | |
| | U | Z | p-value | August n | September n |
| Mean pH$_T$ | 241 | 0.41 | 0.68 | 20 | 26 |
| Min. pH$_T$ | 246 | -0.30 | 0.76 | 20 | 26 |
| Max. pH$_T$ | 212 | 1.05 | 0.29 | 20 | 26 |
| ΔpH$_T$ | 215 | 0.99 | 0.32 | 20 | 26 |
| Mean DIC | 253 | 0.14 | 0.89 | 20 | 26 |
| Min. DIC | 228 | -0.70 | 0.49 | 20 | 26 |
| Max. DIC | 203 | 1.25 | 0.21 | 20 | 26 |
| ΔDIC | 197 | 1.38 | 0.17 | 20 | 26 |
| Mean [CO$_3^{2-}$] | 172 | 1.94 | 0.05 | 20 | 26 |
| Min. [CO$_3^{2-}$] | 225 | 0.76 | 0.44 | 20 | 26 |
| Max. [CO$_3^{2-}$] | 151 | 2.40 | 0.02 | 20 | 26 |
| Δ[CO$_3^{2-}$] | 181 | 1.74 | 0.08 | 20 | 26 |
| Mean Ω$_{arag}$ | 157 | 2.27 | 0.02 | 20 | 26 |
| Min. Ω$_{arag}$ | 213 | 1.03 | 0.30 | 20 | 26 |
| Max. Ω$_{arag}$ | 149 | 2.45 | 0.01 | 20 | 26 |
| Δ Ω$_{arag}$ | 176 | 1.85 | 0.06 | 20 | 26 |
| time Ω$_{arag}$ <1 | 175 | -1.87 | 0.06 | 20 | 26 |
| Mean Ω$_{calc}$ | 168 | 2.03 | 0.04 | 20 | 26 |
| Min. Ω$_{calc}$ | 214 | 1.01 | 0.31 | 20 | 26 |
| Max. Ω$_{calc}$ | 152 | 2.38 | 0.02 | 20 | 26 |
| Δ Ω$_{calc}$ | 178 | 1.81 | 0.07 | 20 | 26 |
| time Ω$_{calc}$ <1 | 218 | -0.9 | 0.4 | 20 | 26 |


## **Acknowledgements**

The authors would like to thank particularly Dr. Todd Martz and his team from the Scripps
Institution of Oceanography (San Diego, USA) for providing the salinity / [O$_2$] / T° sensor package
and for their precious technical assistance. We also wish to thank the Kiel Marine Organism
Culture Centre (KIMOCC) of the Kiel Cluster of excellence "Future Ocean"; Sebastian Fessler
from GEOMAR for the TA / DIC sample measurements; Dr. Bernd Schneider and Stefan Bücker






from the IOW are thanked for valuable discussions with respect to carbonate system analysis in
the Baltic Sea as well as for the inter comparison between TA / DIC / pH measurements; Prof.
Peter Herman from the NIOZ Royal Netherlands Institute for Sea Research for providing access
to the nutrient analyzer; Dr. Jörn Thomsen for his comments on the manuscript; the research divers
from GEOMAR among which are Prof. Martin Wahl, Dr. Christian Pansch, Dr. Yvonne Sawall
and Christian Lieberum; the personnel of the "Seebar am Seebad Düsternbrook" restaurant for
providing electricity to the sensors as well as the Kiel early morning nudist club for its cooperation
in providing access to the Seebar. This work has been funded by the Kiel Cluster of excellence
"Future Ocean".

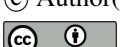



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





Fig. 1. A. Map of the inner Kiel Bay with study site and the GEOMAR station. B. Photo of the sensor suite at the measurement site.

Fig. 2. Deviation between observed and calculated $pCO_2$ ($\Delta pCO_2$) as a function of observed $pCO_2$ illustrating the influence of $TA_{org}$ on carbonate system determinations. Dashed lines represent linear regressions of the respective data. **Black triangles:** $\Delta pCO_2 = pCO_2$ sensor - $pCO_2$ (TA, DIC) for the benthic seagrass deployment of this study, n = 30. The $\Delta pCO_2$ can be explained by a combination of the influence of an organic TA contribution of 0.49 ± 1.47% with measurement uncertainties and sampling errors (spatio-temporal mismatches). **Dark grey diamonds:** $\Delta pCO_2 = pCO_2$ sensor - $pCO_2$ (TA, DIC) based on corrected HydroC® measurements and discrete samples taken at GEOMAR pier in 2015 (Hiebenthal et al., 2017). DIC, TA and $pH_T$ were measured in the same water samples. The $\Delta pCO_2$ is due to a $TA_{org}$ fraction of 0.84 ± 0.0005% (mean ± SD). **Grey circles:** $\Delta pCO_2 = pCO_2$ sensor - $pCO_2$ (DIC, $pH_T$), based on corrected sensor data and discrete samples taken at GEOMAR pier during the same period in 2015, n =18 (Hiebenthal et al., 2017). There is no $pCO_2$ dependency of the $\Delta pCO_2$, because neither DIC nor $pH_T$ are impacted by $TA_{org}$, $R^2 = 0.005$. **Solid black line**: Difference between observed and calculated $pCO_2$ from a model including $TA_{org}$ contributions typical for Baltic Sea water (for details see Appendix B).

Fig. 3. Time series for observed parameters: A: Salinity (black) and temperature (grey), 10 min sampling interval B: Total phosphate (green) and silicate (red) concentration from 31 discrete sampling events at the sensor location. C: $pCO_2$ at 1-min measurement interval (solid line) and 24h moving average (dashed line). D: Dissolved oxygen concentration at 10-min measurement interval (solid green line) and as 24h moving average (dashed green line). Also shown is the oxygen saturation concentration calculated from the water temperature and salinity (black). The straight solid red line represents the hypoxia threshold of 60 µmol kg$^{-1}$.

Fig. 4. Alkalinity (TA) time series (black line) modeled from salinity using the regression equations from August and September. Grey lines represent the $TA_{inorg}$ time series after subtraction of the organic alkalinity contribution of 8 and 30 µmol kg$^{-1}$ from TA, respectively. Triangles (August) and dots (September) represent TA of 31 discrete samples used for the regressions. The sample from the 1st of September at 6:40 (open dot) was not considered for the regression. Insert panel: Linear regressions of TA as function of salinity in discrete samples for the months of August



in blue (n = 14) and September in red (n= 16), see Table 1 for equations and statistics. Dashed
lines are 90% confidence intervals.
Fig. 5. Hourly means (○) ± standard deviation (-) for the months of August (grey) and September
(black) for Temperature (A.), $p$CO$_2$ (B.) and [O$_2$] (C.). Hourly mean ranges and maximum upper
as well as minimum lower standard deviations (-) calculated using TA$_{org}$ contributions of 8 and30
µmol kg$^{-1}$ for pH$_T$ (D.), DIC (E.), [CO$_3^{2-}$] (F.), Ω$_{arag}$ (G.) and Ω$_{calc}$ (H.). The dissolution thresholds
of the Ω values are depicted as solid red lines in panel G. and H.
Fig. 6. Derived time series of carbonate system parameters: pH$_T$ (A),.Dissolved inorganic carbon
(DIC, B.) and saturation states for aragonite (Ω$_{arag}$, light brown) and calcite (Ω$_{calc}$, dark brown)
with dissolution threshold Ω = 1 (red line, C.). All time series are calculated from $p$CO$_2$ (10 min
interval) and total alkalinity with a mean organic contribution of 19 µmol kg$^{-1}$.
Fig. 7. Carbonate ion concentration, [CO$_3^{2-}$], calculated from $p$CO$_2$ (10 min interval) and total
alkalinity with a mean organic contribution of 19 µmol kg$^{-1}$. The red line represents a side specific
[CO$_3^{2-}$] threshold of 80 µmol kg$^{-1}$ below which calcification declines in mussels according to
Thomsen et al., 2015.










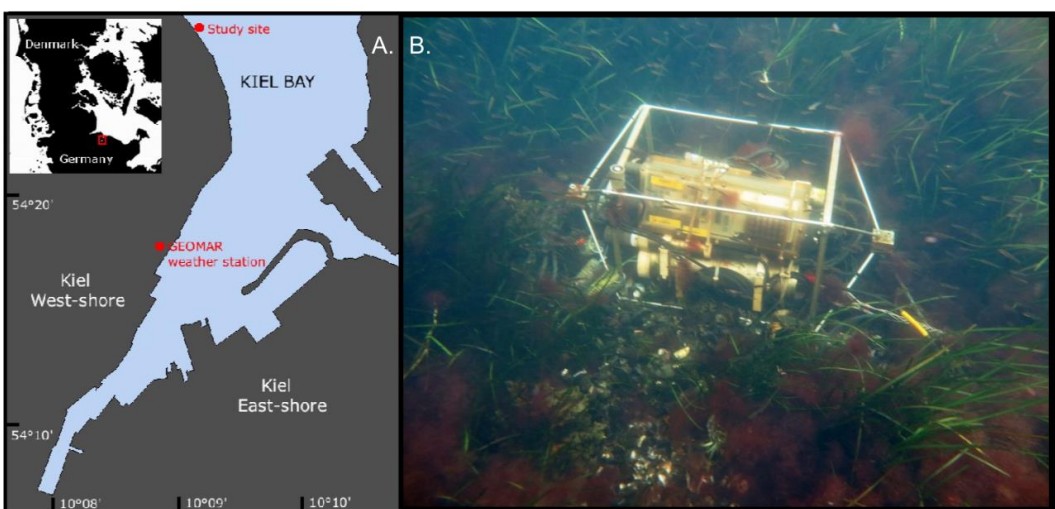





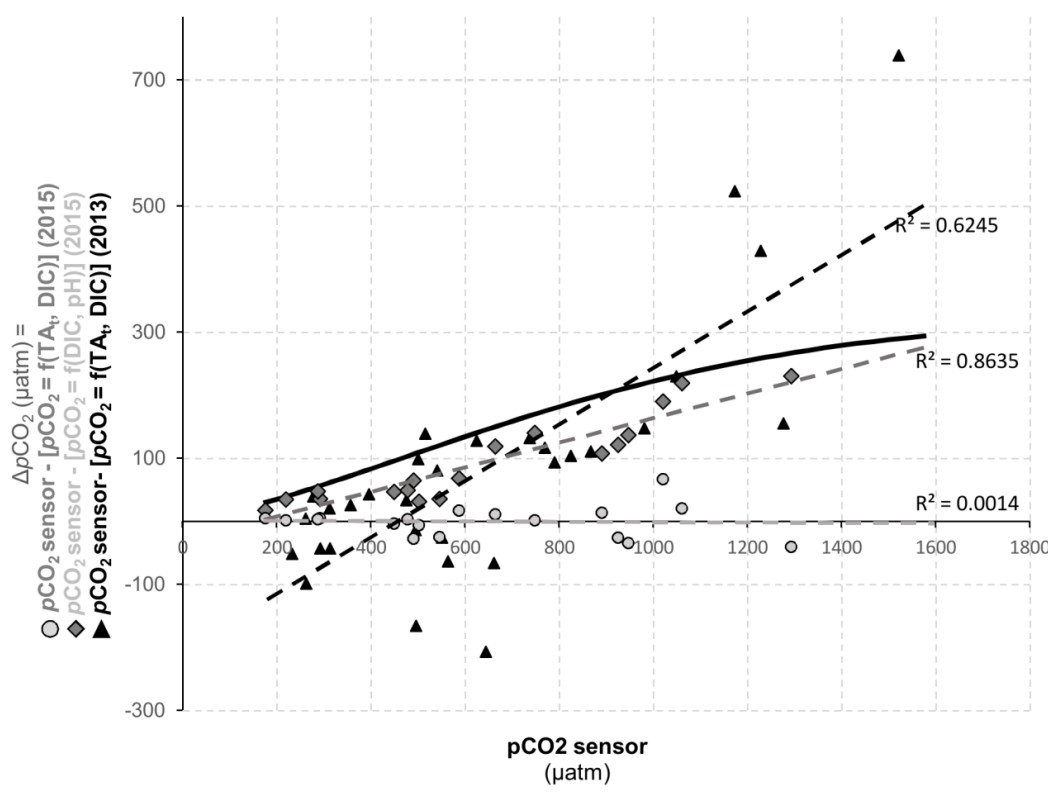


















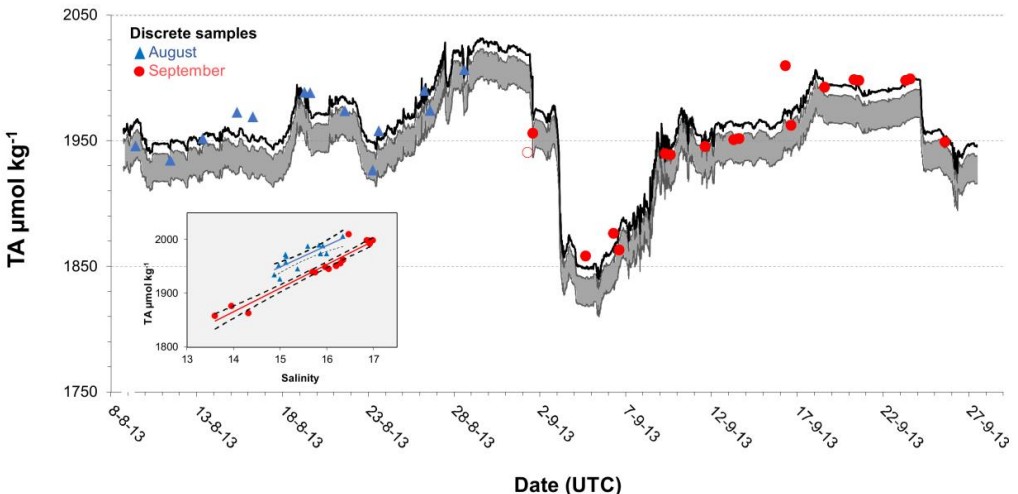






























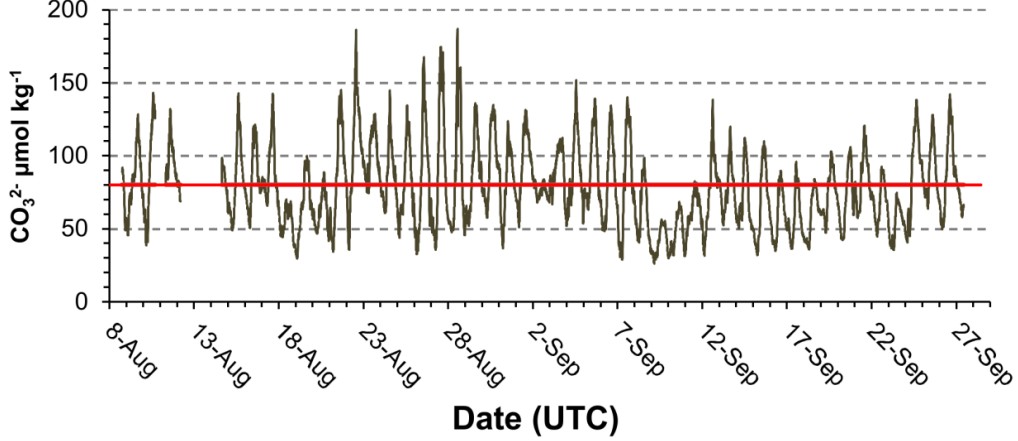
