# Peer review of "Intense pCO2 and [O2] Oscillations in a Mussel-Seagrass Habitat"

_Biogeosciences, 2017_

## Referee Comment (RC1) · Anonymous Referee #1 · 3 Nov 2017

Summary:

Saderne et al. present pCO2 and O2 measurements from a seagrass meadow in Kiel Bay in order to gain insight into coastal carbonate system variability with implications for marine calcifiers (especially the blue mussel Mytilus edulis). They present high-resolution measurements from 7 weeks in the summer of 2013 and use these measurements to calculate other carbonate system parameters of interest. Among this paper's strengths are its error analysis on the measurements from the autonomous pCO2 sensor and its explicit treatment of the role of organic alkalinity in effecting carbonate system calculations. However, I also have some serious reservations about

the carbonate system calculations, especially the result that the mussels experience aragonite undersaturation nearly every day for 6-9 hrs. If this is the case, how do the mussels survive? The fact that the photo of the instruments during their deployment clearly shows mussels suggests that the authors need to either re-visit their carbonate system calculations and/or more critically consider the question they posed in the paper's title, "what are the implications of these measurements for calcifiers?".

This paper has the potential to contribute to the scientific literature on carbonate system variability in nearshore seagrass habitats, but must address some serious shortcomings before I can recommend that it be published.

Below are some general comments for specific sections followed by a list of grammatical and technical comments.

General Comments:

Section 2.2: I have read and re-read your description for the procedures to recover data during the time interval after pCO2 re-zeroing (lines 127-131 + Appendix A). I am still unclear about how you fit your first order kinetics model to the data. I think a simple schematic here would greater improve reader comprehension. I think you do a really nice job with the error analysis here by partitioning the error into the instrumental precision (1%) + the response time uncertainty (1.5%). You add the two components up for a stated uncertainty of 2.5%. However this is only accurate if the errors in the instrumental precision are independent of those from the response time. Have you tested this? If they are not independent, the error will be larger.

Section 2.4: (Please remove the "2.4.1" from the section heading since there is no section "2.4.2" in the paper). I found this section interesting. Given that your model for TA_org plays such an important role in this section (and that the manuscript is not very long in its current form), I think you need to move the full description of your TA_org model from Appendix B and into this section of the main paper. In doing so, I think you will need to include a much more complete error analysis in your model. In Appendix

[Figure]

B, you state that the model provides "only a qualitative approximation of the carbonate system in high DOC waters." Why are you using the model quantitatively then (Fig. 2)? A proper error analysis on the terms of this model would result in some error bounds in addition to the solid black line on Fig. 2. I also do not understand how your calculation of a TA_org contribution of 0.84% from your 2015 bottle samples results in an 8-30 umol/kg contribution range (since 8 umol/kg out of 2000 umol/kg amounts to an error of 0.4% and 30 umol/kg out of 2000 umol/kg amounts to an error of 1.5%). You need to further explain how you arrived at this result. Also, what would it mean for all of your TA_inorg calculations if you adopted the 2013 TA_org contribution value of 0.48% as opposed to the 2015 value of 0.84%.

Section 2.5: This needs to move ahead of Section 2.4. Otherwise, readers do not understand which TA time series you are correcting for the TA_org contribution.

Section 2.6: This does not seem critical to the comprehension of the paper. I recommend relegating this section to an appendix.

Section 2.7: Why are you using Mann-Whitney U tests? What are these tests telling you? Are the samples independent? If not (and I suspect the time series will be highly auto-correlated), these statistical tests are likely inappropriate.

Section 3.2: It would be nice to see the daily means, medians, maximums, minimums, and daily ranges presented as distributions (histograms) that would allow readers to assess measures of central tendency and variance. If your distributions are not normally distributed, reporting your variance as standard deviations is inappropriate.

Section 3.3: Given your significant sources of uncertainty in your TA-S relationship, as well as the uncertainty regarding the contribution of TA_org to the carbonate system calculations, I am very skeptical that your observations of aragonite undersaturation for 6-9 hrs per day are accurate if mussels are living in the seagrass meadow (as you show in Fig. 1B). Overall, I think you need a full error propagation (e.g. Monte Carlo) through all of your carbonate system calculations (including the TA-S relationship) to

better bound your (and the reader's) confidence in your aragonite and calcite saturation state numbers. I would like to see you cross-check your numbers by using the oxygen time series and your DIC bottle samples to calculate a net photosynthetic quotient, which, along with some additional information about O2 advective fluxes, could help you recreate a DIC time series. You could then use this DIC time series (along with its associated errors) + your pCO2 observations to re-calculate aragonite and calcite saturation states. Would they agree with your TA-derived numbers?

Section 4: Your deployment photo clearly shows that your instruments are deployed in the boundary layer, where chemistry can deviate substantially from bulk water conditions. I think this is an important point to consider, as well as an opportunity to highlight the need to co-locate instruments at appropriate depths in the water column if you are trying to infer biological implications from the chemical measurements. Separately, your use of 60 umol/kg as a hypoxic threshold is a very crude measure of hypoxic stress. What does the literature say about hypoxic thresholds appropriate for Mytilus edulis? Overall, I think you need to dive much deeper into the implications of your measurements for mussels and to discuss the next steps for integrating the insights from your chemical measurements into an understanding of how the mussels survive and/or thrive in these conditions. Upon completing the Discussion section, I still have a feeling of "so what?", meaning that I feel unsatisfied in your attempts to link the chemical measurements with an improved understanding of coastal calcifier responses.

Technical Comments:

Line 74: What do you mean by "extension"? Do you mean that it was a circular patch with a radius of less than 2 meters? Or something else? Either way, please clarify

Lines 90-97: Do you have any information about the pCO2 of incoming riverine and/or upwelled water that you could include which would help readers better understand the advective drivers of pCO2 variability in Kiel Bay?

Line 109: What about details on the calibration and precision of your oxygen mea-

surements? You do a great job discussing the quality control and data processing on your pCO2 measurements, but I don't see any additional info for your oxygen measurements.

Lines 151-157: Precision needs to be defined as some recognized statistical measure of variance (e.g. standard deviation, standard error of the mean, etc.).

Line 327: Suggest rewording "However, we note that at not point of our survey the threshold of hypoxia..." to "However we did not observe dissolved oxygen concentrations below the hypoxic threshold during our survey..."

Line 330: "shouts" should be "shoots"

Line 331: replace "auto-" with "autotrophic"

Fig. 2: The y-axis label color scheme (grey circles) does not match the Fig. 2 caption. Please correct.

Fig. 3: I would like to see a plot of O2 percent saturation below the O2 concentration plot

Figs. 6 and 7: Why are these separated into two figures? If no convincing reason, combine into 1 figure.

---

## Referee Comment (RC2) · Anonymous Referee #2 · 26 Feb 2018

Summary: This study reports pCO2, O2, salinity, and temperature sensor time-series in August and September 2013 in Kiel Bay. Sensors were deployed in seagrass mussel habitat. Using discrete samples of TA, DIC, phosphate, and silicate, the authors calculate calcite and aragonite saturation states. The authors placed a lot of emphasis on contributions of organic TA to these calculations, despite the fact that they concluded that the contribution was low. In the Discussion, the authors briefly relate the conditions to regional physiological tolerance of Mytilus edulis.

General comments

Overall, I think that the time-series data is of publishable quality, pending additional

reporting of accuracy and uncertainty associated with the carbonate system calculations (by incorporating the uncertainty of the measurements - this can be done in the R Seacarb package, see comments below). These kind of data are necessary, however, the time-series too short to fully describe what organisms experience in this habitat. My main concerns relate to how the data are used to generate a manuscript:

(1) There is no question or hypothesis that is being tested for the given (and incorrect) statistical approach. Comparison of two months is an odd approach for high-frequency time-series of two months. Why are August and September being compared so distinctively? Is there some important biological process that occurs during these months? It's time-series data, and the data are not independent (required for the Mann-Whitney U test). The authors do not spend much time on the one interesting event in this time-series: TA and declined in early August, altering the TA-salinity relationship.

(2) The emphasis on TAorg appears very small, yet such big emphasis is placed on trying to identify TAorg contributions. The authors do not adhere to one method in terms of assessing TAorg in their study (see details below). This is confusing and, as currently presented, makes it seem like the authors cannot decide how to integrate TAorg in their study.

As presented, this dataset makes for a weak standalone paper. It seems that this dataset should be paired with a biological study on the mussels (as indicated by the title). I assume, given the motivation for measurements at this site, that this is underway. A sensor-biology pairing will be a much more powerful than a stand-alone chemistry and biology paper.

Specific comments on TAorganic

If TAorg contribution is low in this region (as they authors conclude, L300-304), the authors should just state that. If there is no citation, an appendix could be added explaining why. TAorg was not the focus of this study, it's deemed not important, yet it takes up a large portion of the manuscript (but ignored in the Discussion). The authors

present various estimates for TAorg and do not adhere to one approach for use of TAorg in their dataset. Instead, they just present all options. If I somehow misunderstood the intent, I urge the authors to greatly clarify their intention and relevance of doing so.

L187: I don't understand the relevance of the modeling approach in the context of the small spatial context of this study. The data from this study already show the increasing offset with increasing sensor pCO2.

In Fig. 2, I don't understand why one dataset from GEOMAR pier contains no TAorg dependency and the other one does (grey circles vs dark grey diamonds). Are the TA fraction results from Hiebenthal et al? If the TAorg contribution at GEOMAR varies, that is further justification to not blindly apply the TAorg contribution from GEOMAR to the current study site.

L205-213: I don't understand why the authors are not confident in their own bottle sample data and instead choose to use data from a different site from a different year to estimate the contribution of TAorg to TA during their study. Surely a spatio-temporal mismatch error at the study site in 2013 would be smaller than the error introduced by using data from GEOMAR in 2015.

L209: How was the 0.48% error calculated (it is also different from what is reported in Fig. 2 legend. Is this the same calculation)? Is the sensor data from Fig. 2 grey circles all from 2013? This section, as well as the Fig. 2 legend is confusing.

3.1: Why is the salinity derived TA time-series being reported on after all the focus on TAorg contributions?

3.3: Why are timeseries calculated using TAinorg but then analyzed with TAorg of 8 and 30 in Table 1? Why and when is TAorg 19 used in Fig 7 and where does that number come from?

4. Discussion: no mention of TAorg.

Specific comments:

Title: "Implications for calcification" is misleading, as no biological data is included.

L26: perhaps report the full range of diel pH cycles (min and max) rather than the SD, which is quite large.

L37-38: report CO2 concentration in the same units.

L64: include upwelling

L102: explain why this site was chosen

L113: is this a SeapHOX? Is there a reason why the pH data is not included?

L127: specify which sensor this paragraph relates to

L139: Error in pCO2 is 2.5% (although the authors should clarify what they mean by standard deviation - are they calling the error a standard deviation?). Given the emphasis on saturation states, this error and those of the bottle samples should be extrapolated and saturation states should be reported with an error (graphically).

L147-149: explain why these sampling times were chosen. Was this every third day or only once? Which corresponding samples are averaged? Since this is a dynamic environment, averaging measurements taken one hour apart may not be appropriate. Please justify and report agreement of these samples.

L150: samples were preserved after salinity measurements? That implies CO2 off-gassing. I assume that the preservation was done immediately. Please clarify.

L152-154: need to report accuracy, in addition to the precision. Describe how accuracy was determined.

L155-158: need to report accuracy, in addition to the precision. Describe how accuracy was determined.

L176: specify that this sentence is based on data from Kiel Fjord (reads as if it is the Baltic)

[Figure]

L308: Discuss why the CO2 baseline is different between the current study and Saderne et al. 2013.

L352: Frieder uses semidiurnal cycles, not diel

L361: I don't think this study demonstrates this point

L365: Again, TAorg was deemed not important, so why is its contribution emphasized here?

L369: Why is this site exemplary? The site choice was not justified anywhere in the text.

Fig. 1A: add scale bar

Fig. 3C: add atmospheric CO2 time-series to this plot (given Sect. 3.2)